# Implications of Hereditary Origin on the Immune Phenotype of Mismatch Repair-Deficient Cancers: Systematic Literature Review

**DOI:** 10.3390/jcm9061741

**Published:** 2020-06-04

**Authors:** Lena Bohaumilitzky, Magnus von Knebel Doeberitz, Matthias Kloor, Aysel Ahadova

**Affiliations:** 1Department of Applied Tumor Biology, Institute of Pathology, University Hospital Heidelberg, 69120 Heidelberg, Germany; l.bohaumilitzky@dkfz-heidelberg.de (L.B.); magnus.knebel-doeberitz@med.uni-heidelberg.de (M.v.K.D.); Matthias.Kloor@med.uni-heidelberg.de (M.K.); 2Clinical Cooperation Unit Applied Tumor Biology, DKFZ, 69120 Heidelberg, Germany; 3Molecular Medicine Partnership Unit (MMPU), University Hospital Heidelberg, 69120 Heidelberg, Germany

**Keywords:** colorectal cancer, DNA mismatch repair deficiency, immune evasion, immune infiltration, Lynch syndrome, microsatellite instability

## Abstract

Microsatellite instability (MSI) represents one of the major types of genomic instability in human cancers and is most common in colorectal cancer (CRC) and endometrial cancer (EC). MSI develops as a consequence of DNA mismatch repair (MMR) deficiency, which can occur sporadically or in the context of Lynch syndrome (LS), the most common inherited tumor syndrome. MMR deficiency triggers the accumulation of high numbers of somatic mutations in the affected cells, mostly indel mutations at microsatellite sequences. MSI tumors are among the most immunogenic human tumors and are often characterized by pronounced local immune responses. However, so far, little is known about immunological differences between sporadic and hereditary MSI tumors. Therefore, a systematic literature search was conducted to comprehensively collect data on the differences in local T cell infiltration and immune evasion mechanisms between sporadic and LS-associated MSI tumors. The vast majority of collected studies were focusing on CRC and EC. Generally, more pronounced T cell infiltration and a higher frequency of *B2M* mutations were reported for LS-associated compared to sporadic MSI tumors. In addition, phenotypic features associated with enhanced lymphocyte recruitment were reported to be specifically associated with hereditary MSI CRCs. The quantitative and qualitative differences clearly indicate a distinct biology of sporadic and hereditary MSI tumors. Clinically, these findings underline the need for differentiating sporadic and hereditary tumors in basic science studies and clinical trials, including trials evaluating immune checkpoint blockade therapy in MSI tumors.

## 1. Introduction

DNA mismatch repair (MMR) deficiency is a major pathway enabling genomic instability in human cancers. MMR deficiency is characteristic for 15% of colorectal cancers (CRC) and 30% of endometrial cancers (EC) (Figure 1a) [1]. MMR deficiency is caused by the inactivation of one of the MMR genes (*MLH1*, *MSH2*, *MSH6* and *PMS2*). As a consequence of non-functional MMR, tumors accumulate insertion/deletion mutations at repetitive microsatellite sequence stretches and therefore present with the microsatellite instability (MSI) phenotype [2].

Indel mutations at coding microsatellites (cMS) inactivate critical tumor suppressor genes and promote tumor development. Simultaneously, they can trigger shifts of the translational reading frame and lead to the generation of frameshift peptides (FSPs). FSPs play a major role in the pronounced immunogenicity of MSI CRCs as they often encompass long neopeptide stretches and multiple potential epitopes that can be recognized by the immune system (Figure 1b). FSPs are not only tumor-specific neoantigens but also shared by most MSI tumors, as they are the result of certain driver mutations that are positively selected during tumor evolution, e.g., the cMS tract in exon 3 of the *TGFBR2* (transforming growth factor beta receptor 2) gene is mutated in around 90% of MSI CRCs [2]. The immunogenicity of certain FSPs was comprehensively demonstrated *in vitro* [3,4,5].

Likely because of their high load of neoantigens, MSI cancers, typically those in the colorectum, display signs of high immunogenicity, such as pronounced local T cell infiltration [6,7,8,9], elevated counts of tumor-infiltrating lymphocytes (TILs) with cytotoxic potential [9,10,11] and a high frequency of immune evasion phenomena [2,12,13]. Patients with MSI CRCs also show a more favorable prognosis in a stage-wise comparison with microsatellite-stable (MSS) tumor patients, potentially reflecting active anti-tumor immune responses [14,15]. Notably, treatments supporting the anti-tumoral immune response, such as the recently developed immune checkpoint blockade (ICB) therapy, showed great success specifically in patients with metastasized MSI cancers [16,17].

Although these clinical and immunological observations have been documented by several independent studies, little attention has been attributed to the influence of the origin of MSI cancers on their immunogenicity: Whereas the majority of MSI CRCs develop sporadically, mostly due to hypermethylation of the *MLH1* promoter leading to *MLH1* silencing [18,19,20,21,22], about 20–30% of MSI CRCs (Figure 1a) have a hereditary background and are associated with Lynch syndrome (LS), the most common inherited CRC syndrome [23]. LS is caused by an inherited monoallelic germline alteration inactivating one of the MMR gene alleles and leading, upon a second somatic hit, to the molecular phenotype of MMR deficiency in the affected cells [24,25].

The overall lifetime cancer risk in LS depends on the affected MMR gene and is estimated to vary between 50% and 80% for the high penetrance genes, *MLH1* and *MSH2* [26,27,28,29,30,31,32]. Clinically, LS CRC patients typically present with cancer onset at a younger age (∼45 years) compared to sporadic MSI CRC patients [27]. Due to the high lifetime cancer risk associated with LS, carriers often, but not always, fulfill clinical criteria indicative of hereditary cancer such as young age of onset or multiple cancer manifestations (Bethesda criteria) [33,34].

There are profound differences in the pathogenesis of sporadic and hereditary MSI CRCs. Whereas sporadic MSI tumors predominantly arise through *BRAF*-mutated serrated lesions [35,36], LS-associated CRCs have been described to arise either through conventional adenomas [37,38,39] or through MMR-deficient crypt foci (MMR-DCF) [40,41,42,43]. MMR-DCF lack the expression of MMR proteins but, in contrast to adenomas, are morphologically undistinguishable from normal colonic crypts. MMR-DCF are specifically associated with LS and found in abundance in the intestinal mucosa of LS carriers (about 1 per 1 cm^2^) [41]. Recently, evidence for the true precancerous nature of MMR-DCF has emerged, including the molecular observation of MSI and mutations in cMS (e.g., *HT001*, *AIM2*, *BA*X, *TGFBR2*) in larger MMR-DCF [41,42]. The ongoing, FSP-associated immune surveillance in LS mutation carriers would potentially enable the elimination of most MMR-deficient cell clones.

Based on the differences between LS and sporadic MSI tumorigenesis, it seems plausible that the distinct evolutionary forces shaping the tumors are also reflected in phenotypic differences. However, evidence supporting this hypothesis has so far been scarce and scattered across the scientific literature. To the best of our knowledge, no comprehensive literature analysis comparing immune characteristics of LS and sporadic MSI tumors exists yet. In the present systematic literature review, we aim to comprehensively describe the local immune phenotype of MSI tumors, with emphasis on differences between LS-associated and sporadic MSI tumors. The existence of such differences would provide new insights into the role of immune surveillance in LS-associated and sporadic MSI cancers and guide therapeutic approaches in these molecularly distinct tumor types.

## 2. Methods

### 2.1. Systematic Literature Search

A systematic literature search was conducted to identify studies that compared the extent of immune infiltration and immune evasion in sporadic and hereditary MSI CRCs. The inclusion criteria were direct comparison of immune characteristics between sporadic and LS-associated MSI tumors, irrespective of tumor location and evaluation method. Exclusion criteria were comparison between MSS and MSI only, not considering hereditary or sporadic pathogenesis of MSI tumors, as well as lack of clear definition of the comparison groups. 

An online MEDLINE search (http://www.pubmed.com) was conducted between 29 February and 10 March, 2020 and the following keywords were used: {Lynch} OR {Lynch syndrome} OR {HNPCC} OR {MSI} OR {microsatellite instability} OR {MMR-deficient} OR {MMR-deficiency} OR {hereditary MSI} OR {sporadic MSI} OR {microsatellite instable} OR {RER +} OR {mutator phenotype} AND {immune infiltration} OR {immune microenvironment} OR {T cell infiltration} OR {TIL} OR {tumor infiltration} OR {immune evasion} OR {T cell density} OR {B2M} OR {RFX5} OR {CIITA} OR {HLA} OR {TAP} OR {NLRC5}.

The screening of retrieved literature was performed manually. In addition, search of studies cited in the manuscripts that were analyzed on the full text level was performed. Although sample overlap between some of the analyzed studies cannot be formally excluded, sample collection from different sources or substantial temporal separation/analysis of different immune markers in the studies with mutual sources argues against major influence of the sample overlap on the results. A subset of the original search results was analyzed by two observers independently. The inter-rater reliability was measured by the Cohen’s kappa coefficient using Rstudio [44]. The strength of agreement on the title level was found to be substantial (*κ* = 0.7). The status of the few studies with discrepant evaluation was unanimously agreed on by both observers when abstracts were analyzed. 

Identified studies were checked for their eligibility by screening title, abstract and, lastly, the full text article. During the screening process, Preferred Reporting Items for Systematic Reviews and Meta-Analyses (PRISMA) recommendations were followed (Figure 2). The selected papers were analyzed qualitatively and quantitatively whenever possible. Quantitative analysis of the immune infiltration data was not conducted as different antibodies and counting methods (percentage of positive staining vs. cell counts) were used, and different cell categories (e.g., stromal vs. epithelial) were assessed. Additional studies and review articles addressing the immune phenotype of MSI CRCs were identified manually and independently of the used search terms from the authors’ own files.

### 2.2. Statistical Analysis 

All statistical evaluations were performed in GraphPad Prism (Version 6.07, GraphPad Software Inc., La Jolla, CA USA) and Rstudio (Version 3.6.1, RStudio Inc., Boston, MA USA) [44]. The 95% confidence interval (CI) of the proportion of *B2M* mutations in the hereditary and sporadic MSI CRC groups was calculated with the modified Wald method in GraphPad Prism. Fisher’s exact test (*α* = 0.05) was used to check for a statistically significant association between *B2M* mutations and the origin of the MSI CRC; the analysis was also performed in GraphPad Prism. Odds Ratios (OR) and respective 95% CIs for the hereditary vs. the sporadic MSI CRC group were calculated and illustrated using the Rstudio packages epitools [45] and ggplot2 [46] Subsequently, statistical significance was checked using a Fisher’s exact test (*α* = 0.05).

## 3. Results

### 3.1. Outcome of the Systematic Literature Search

The literature search with the defined keywords covering immune infiltration and immune evasion (see Methods) revealed 522 articles in total. Following PRISMA recommendations, all articles (*n* = 522) were screened, and 364 publications were excluded based on the title. From the remaining 158 articles, 88 were excluded based on the abstract and 49 were excluded on the basis of the full text (Appendix A). 17 eligible articles fulfilled the criteria and were included in the final analysis. The selection process is visualized in Figure 2. The majority of identified studies focused on MSI CRC, whereas only a small proportion of studies analyzed MSI EC.

### 3.2. Immune Infiltration in Hereditary and Sporadic MSI CRCs

We first compared the immune infiltration determined by the quantitative analysis of different immune markers (CD3, CD4, CD8, PD1, PD-L1, FoxP3, CD45) between hereditary and sporadic MSI cancers. Seven studies [47,48,49,50,51,52,53] were informative for quantitatively evaluating the extent of local immune infiltration in MSI CRCs (Table 1). We looked at the results from the immunohistochemical analysis of different T cell subsets of these studies, where a total of 122 hereditary MSI CRC and 145 sporadic MSI CRC samples were examined. A direct comparison between hereditary and sporadic MSI CRC groups was performed in each study. The remaining ten studies that did not provide exact T cell counts but worked with frequencies or defined categories are not included in Table 1 but will be discussed below.

Studies using the pan-T cell marker CD3 for assessing intraepithelial T cell infiltration in sporadic and hereditary MSI CRCs uniformly observed higher T cell counts in the hereditary group [47,48]. Equivalently, Young et al. (2001) reported a higher frequency of T cell infiltration in the hereditary group (73%) compared to the sporadic MSI CRC group (56%) [54]. Further, a study by Jass et al. (2002), which was identified independently of the literature search, also mentions a higher frequency of Crohn’s like reactions and peritumoral T cells in hereditary compared to sporadic MSI CRCs [55]. However, Takemoto et al., who did not use exact numbers for intraepithelial T cells but instead distinguished between poor, moderate and severe intraepithelial T cell infiltration, found no significant difference between the two groups [49]. Similarly, Wang et al., using hematoxylin and eosin slides to quantify T cell infiltration, did not find a significant difference between sporadic and hereditary MSI CRCs [56].

In addition to pan-T cell counts, some studies differentiated between T cell subpopulations. For instance, Takemoto et al. reported slightly higher numbers of CD8- and CD4-positive T cells in the stroma of hereditary compared to sporadic MSI tumors [49]. However, other studies observed similar CD8 and CD4 infiltration levels in both MSI CRC groups [50,51,57].

### 3.3. Immune Infiltration in Hereditary and Sporadic Pre-malignant Colorectal Lesions

In summary, the existing literature provides ample evidence that LS-associated MSI CRCs generally have a more pronounced local T cell infiltration compared to sporadic MSI CRCs (Figure 3). Although the evidence level is lower and further research is warranted, enhanced immune cell reactivity in LS-associated lesions seems to be already present at a premalignant stage.

Our literature search identified a few studies analyzing immune infiltration not only in manifest cancers, but also in cancer precursors. Koornstra et al. (2009) reported significantly higher CD8-positive T cell counts in LS-associated compared to sporadic adenomas; however, the MSI status of these adenomas was not specified. Notably, T cell infiltration of LS-associated adenomas was significantly lower than the T cell infiltration of LS-associated CRCs [50]. Fittingly, another study [58] reported a higher frequency of T cell infiltration in LS-associated, MMR-deficient adenomas, compared to sporadic adenomas. However, as sporadic adenomas are mostly MMR-proficient, one should be careful when comparing LS-associated adenomas to this control group. An indication for activation of the immune system in premalignant lesions of LS patients has also been reported by Chang et al. who found increased RNA expression levels of CD4, IFNG, LAG3, PDL1, IL12A and TNF in LS-associated compared to Familial Adenomatous Polyposis (FAP)-associated adenomas suggesting immune activation in LS-associated adenomas. Interestingly, the authors reported that this immune profile was not related to the MMR status and mutational rates [59]. Further experiments are required to validate these findings and to provide visualization of immune cells infiltrating LS-associated adenomas.

Alongside with the direct assessment of T cell infiltration, the immune status of a lesion can be elucidated from other indirect markers of immune response, such as expression of immune checkpoint molecules or lymphocyte recruitment markers. The expression of PD-L1, an interferon-gamma-inducible marker of prolonged immune activation, did not display significant differences between LS-associated and sporadic MSI CRCs [60,61]. However, Pfuderer et al. found much higher densities of high endothelial venules (HEV) in peritumoral tissue adjacent to hereditary MSI CRCs than to sporadic MSI CRCs [62]. HEVs are specialized postcapillary venules responsible for lymphocyte trafficking [62,63]. The evidence for enhanced lymphocyte recruitment to LS-associated MSI CRCs may point towards pre-existing immune responses in hereditary MSI patients induced by MMR-DCF [42]. Thus, long before tumor manifestation, MMR-DCF may prime the immune system of LS carriers against specific cMS mutation-induced FSP neoantigens. This is supported by the observation of FSP-specific immune responses in the peripheral blood of tumor-free LS carriers [64]. A primed immune system may facilitate strong immune infiltration of manifest cancers in the setting of LS but not, or only to a lesser degree, in sporadic MSI cancers.

### 3.4. Immune Infiltration in Hereditary and Sporadic ECs

The phenomenon of enhanced immune responses associated with the hereditary origin of MSI cancer is not restricted to the colon. A study by Pakish et al. (2017) showed that a trend towards elevated immune cell infiltration can also be observed in LS-associated MSI ECs. The stromal compartments of LS-associated ECs displayed a significantly higher number of CD8-positive T cells, compared to the sporadic controls. In addition, higher CD8-positive intraepithelial T cell counts were observed in LS-associated ECs, but this increase did not reach statistical significance [52]. Similarly, Ramchander et al. reported a significantly elevated number of CD8-positive T cells in the invasive margin of LS-associated MSI ECs compared to sporadic cases. The same tendency was observed in the tumor center but failed to reach statistical significance [53]. Furthermore, another study reported a higher frequency of PD-L1 expression in LS-associated MSI ECs, compared to sporadic MSI ECs [65]. Ultimately, the picture of immune responses in LS-associated and sporadic MSI CRCs may go beyond tumor- and adenoma-related immune infiltration, as systemic immune responses against FSP neoantigens have been detected in tumor-free LS mutation carriers [64]. This finding may point at tumor-independent differences in the immune phenotype between the general population and LS carriers being present long before tumor manifestation and possibly influencing cancer risk in LS carriers. Such systemic immune responses might be linked to LS-associated MMR-deficient lesions, which can be detected not only in colorectal [40,41] but also in the endometrial tissue of LS carriers [66].

### 3.5. Immune Evasion Mechanisms in Hereditary and Sporadic MSI CRCs

Pronounced immune infiltration and strong local immune response can exert a substantial selective pressure on emerging precancerous cell clones [67]. This can trigger immune evasion through positive selection of tumor cell clones capable to escape destruction via the immune system. The main immune evasion mechanism in MSI CRCs is impairment of HLA class I-mediated antigen presentation, present in approximately 70% of all MSI CRCs [13]. In MSI CRCs, HLA class I-mediated antigen presentation is most frequently abrogated by mutations in the *B2M* (Beta-2 microglobulin) gene, which is affected in 30% of MSI CRCs. The *B2M* gene contains four microsatellites in its coding region, which make this gene susceptible for mutation in MMR-deficient cells [12,13,68,69]. Interestingly, the observed frequency of *B2M*-inactivating mutations is significantly higher than expected by chance based on the short length of the microsatellites [70], illustrating the tremendous survival advantage of *B2M*-mutant MSI clones under immune surveillance. Moreover, *B2M* mutations are closely associated with active local immune responses, represented by high numbers of activated T cells and a high density of HEVs [62,71].

Five studies were identified that could be analyzed regarding immune evasion mechanisms affecting HLA class I- and HLA class II-related antigen presentation in hereditary and sporadic MSI CRCs (Table 2).

Loss of HLA class I due to *B2M* mutations was examined in a total of 178 hereditary and 166 sporadic MSI CRCs. The proportion of *B2M*-mutated LS-associated MSI CRCs varied between 17% and 50%. For sporadic MSI CRCs this proportion ranged between 3% and 29%. The majority of included studies reported a higher frequency of *B2M* mutations in hereditary MSI CRCs compared to sporadic cases [12,47,73] (Figure 4). Only one study identified through our literature search [72] reported a higher rate of *B2M* mutations in sporadic compared to LS-associated CRCs. Although the reasons for this discrepancy are not clear, the study by Clendenning et al. confirmed previous reports that *B2M* mutations are associated with increased immune infiltration and elevated numbers of Crohn’s like lesions in the tumor vicinity, supporting the hypothesis of immunoediting as the cause of outgrowth of tumor cells with impaired B2M [72].

In addition, two studies analyzed the frequency of *B2M* mutations in adenomas: Kloor et al. (2007) reported a 15.8% *B2M* mutation frequency in 38 MSI adenomas [12]. In contrast, Clendenning et al. did not observe any *B2M* mutations in 42 MMR-deficient adenomas [72]. Although the hereditary or sporadic origin of the analyzed adenomas was not clearly specified in these two studies, the current literature suggests high likelihood of MSI/MMR-deficient adenomas to be indicative of LS [74].

The frequency of *B2M* mutations also seems to be influenced by the affected MMR gene, as a genomic and transcriptomic characterization of LS-associated CRCs revealed that *MLH1*-mutated tumors, compared to those having defects in the other MMR genes, may present with a higher rate of *B2M* mutations [75]. This observation potentially implies that MLH1-deficient tumors may be exposed to more pronounced immune responses than MSH2-deficient tumors. Similar findings have been reported independently by Clendenning et al. (2018) and Echterdiek et al. (2016) [71,72]. The mechanisms underlying this difference between MLH1- and MSH2-related immune phenotypes are not fully understood yet and require further research.

Further studies are also necessary to evaluate potential differences related to the HLA class II antigen presentation machinery, which may have an important role in MSI cancer immunology [76]. Mutations of components involved in the HLA class II antigen processing and presentation pathway have so far only been addressed by one study [51], which analyzed 35 hereditary and 34 sporadic MSI CRCs for CIITA and RFX5 mutations. Although not reaching statistical significance, potential differences were observed in the patterns of mutations in HLA class II-regulatory genes, as CIITA mutations were more frequent in hereditary MSI CRCs, whereas RFX5 mutations occurred more frequently in sporadic MSI CRCs [51].

### 3.6. Response to Therapy in Hereditary and Sporadic MSI CRCs

Long before the era of ICB treatment, differences in therapy response have been observed between MSI and MSS CRC patients. 5-Fluorouracil (5-FU) treatment demonstrated limited therapeutic effects in MSI CRC patients and has even been reported to be associated with detrimental effects [77,78,79]. Therefore, adjuvant 5-FU-based chemotherapy is not recommended for MSI CRC patients of UICC (Union for International Cancer Control) stage II. However, MSI CRC patients seem to respond well to irinotecan-based therapies [80,81]. The mechanisms responsible for these differences are not fully clear. Besides potential immune modulation by chemotherapy, MMR deficiency may directly influence responsiveness on the cellular level. A functional MMR system may be required for cell death induction after 5-FU integration [80], whereas irinotecan-induced impairment of DNA damage repair may be synthetically lethal in MMR-deficient cells. Only few studies so far analyzed chemotherapy responsiveness differentially between hereditary and sporadic MSI CRCs. Sinicrope et al. (2011) reported a significant reduction of distant recurrences and improved disease-free survival in LS-associated MSI CRC patients compared to sporadic MSI CRC patients, both groups receiving 5-FU-based chemotherapy [82]. In addition, a study by Zaanan et al. reported a significantly improved progression-free survival for patients with LS-associated MSI CRCs under chemotherapy plus anti-EGFR, compared to patients with sporadic MSI CRCs [83].

Recently, ICB therapy revolutionized the field of oncology by showing a tremendous success in metastasized (UICC stage IV) MSI tumors. Blockade of the PD-1 receptor, e.g., with Pembrolizumab, shows success in treating metastatic MSI cancers irrespective of the tissue of origin [16,84,85]. A phase II study conducted by Le et al. demonstrated enhanced responsiveness of MMR-deficient CRCs to Pembrolizumab treatment, in comparison to MSS CRCs [16]. However, little is known about potential differences between sporadic and hereditary MSI CRCs under ICB treatment. Differences between the immune phenotype of sporadic and hereditary MSI cancers would be of immediate clinical relevance for ICB therapy, because sporadic MSI cancers can be differentiated from hereditary MSI cancers by molecular tumor analysis. If one of the two groups showed preferential response to ICB, information about hereditary and sporadic origin may guide patient selection for treatment.

Based on the data reported above, two major implications of pathogenesis on ICB responsiveness are conceivable.On the one hand, LS-associated MSI CRCs, according to the performed literature search, show an elevated density of local immune cells compared to the sporadic group. Local tumor infiltration with cytotoxic T cells has been reported as a prerequisite for ICB response [86,87]. Therefore, LS patients with MSI cancers may have a higher chance of having sufficient numbers of T cells present at the tumor site, which can be reactivated by ICB to attack tumor cells.

On the other hand, most of the studies reported above found a higher rate of impaired HLA class I-mediated antigen presentation, mainly caused by *B2M* mutations, in hereditary MSI CRCs. Therefore, the opposite constellation is reasonable to assume; LS-associated MSI CRCs may more frequently show upfront resistance towards ICB, which however might be masked by the generally favorable prognosis of patients with *B2M*-mutant tumors [12,69]. Although the reason for the latter observation is not yet fully clarified, different theories explaining this counter-intuitive association exist, including enhanced NK cell-mediated tumor killing or lower metastatic capacity due decreased platelet binding [2,88,89,90].

Le et al. found no significant difference in the objective response rate between the hereditary and sporadic patient group receiving PD-1 blockade therapy, possibly because positive and negative factors offset each other [16,17]. The scarce data on differential responsiveness of MSI cancer patients based on hereditary or sporadic pathogenesis clearly indicates that future studies with systematic classification of patients according to the origin of the tumor will be necessary. Moreover, information about immune infiltration and immune evasion phenomena needs to be recorded for tumors treated by ICB.

### 3.7. Alternative Explanations for Differing Immune Phenotypes

In addition to the hypothesis of immune stimulation via FSP neoantigens prior to tumor development in LS setting, other explanation for the observed immunological differences should be considered, such as host factors that shape the immune system and respective immune responses. Recently, the importance of the host’s immune status in the LS setting has been highlighted by a study showing that an immune dysfunction due to immunosuppression or autoimmune conditions is associated with a higher rate of multiple cancers [91]. This association underlines the major role of immune surveillance in determining LS cancer risk and indicates the necessity accounting for the immune status of LS carriers when performing cancer risk estimation studies.

LS-associated MSI cancers have a younger average age of onset than sporadic MSI cancers [27,92]. As the activity of the immune system and probably its capacity to recognize tumor cells are related to age [93,94,95], one may speculate that the observed elevated immune responses in LS mutation carriers compared to sporadic MSI tumor patients may merely reflect the generally younger age of the former. However, the local immune activation reflected in the HEV density and detected specifically in LS-associated tumors remained significantly higher than in sporadic MSI tumors even after correction for age [62], indicating that age alone cannot explain the enhanced local immune responses against LS-associated compared to sporadic MSI CRCs.

Additionally, gender should be taken into account when discussing the differences between hereditary and sporadic MSI CRCs as sporadic MSI CRCs predominantly occur in women. This is also reflected by the gender-specific differences in tumor location, molecular characteristics and mortality of CRCs [96]. Right-sided CRCs occur more frequently in women than in men [97]. In addition, the MSI tumor phenotype is also more common in women and a link to estrogen exposure could be made. Especially older, menopausal women with lowered estrogen levels are at risk to develop sporadic MSI CRCs, and it has been shown that hormone replacement therapy reduces the risk for MSI CRCs [98,99]. Females also present with a higher incidence of *BRAF* mutations associated with sporadic MSI cancers as a consequence of age-related enhanced levels of DNA methylation and silencing of tumor-suppressor genes responsible for oncogene-induced senescence [96,100].

Besides being involved in tumorigenesis, *BRAF* mutations are also associated with immunosuppression [101,102,103,104]. As the *BRAF* V600E mutation very rarely occurs in LS-associated CRCs (1.4%) but is displayed by around 64% of sporadic MSI CRCs with *MLH1* methylation [105], the proposed immunosuppressive features of this mutation have to be considered when comparing local immune responses in the two MSI CRC groups. However, the majority of studies reporting immunosuppressive effects of *BRAF* mutations were done in melanoma and some additional evidence points to the absence of BRAF-mediated immunosuppression in the context of MSI CRCs: First, TIL density is not significantly reduced in *BRAF*-mutated MSI CRCs, compared to *BRAF*-wild-type MSI CRCs [106]. Second, *BRAF*-mutated MSI CRCs responded to ICB treatment with Nivolumab plus Ipilimumab as well as *BRAF*-wild-type MSI CRCs [84]. Third, *BRAF* mutation in the MSI setting is not associated with poorer prognosis and reduced overall survival, in contrast to MSS CRCs [107].

It is worthwhile to note that studies analyzed in this systematic literature review were heterogenous with regard to the definition of LS: whereas some studies only included proven pathogenic MMR variant carriers, others relied on LS definitions based on immunohistochemistry or fragment length analysis suggestive of the MMR-deficient/MSI phenotype, complemented by *MLH1* methylation/*BRAF* mutation analysis and clinical criteria. Omitting such studies would have substantially reduced the comprehensiveness of our survey. As the percentage of tumors wrongly classified using the “suspected LS” classifier is expected to be low based on existing literature data, we decided to include both types of studies. Future studies addressing the immunological differences between sporadic and LS-associated MSI tumors are encouraged to define LS based on pathogenic MMR variants in the germline.

## 4. Summary

Our systematic literature analysis has two major findings: First, MSI cancers developing in the context of LS show more active local immune responses than sporadic MSI cancers. Second, and possibly related to the first finding, immune evasion phenomena in MSI cancers seem to be associated predominantly with a LS background. We found evidence that recurrent encounters with premalignant lesions prime the immune system against MSI-associated tumor antigens in LS individuals (Figure 5). Possibly, elevated immune surveillance is responsible for the incomplete penetrance of LS. However, to verify these hypotheses and gain more confidence in the general character of the findings currently based on CRC and EC, studies analyzing immunological features in other MSI tumors of the LS spectrum are warranted.

Our study demonstrates that classification of a tumor as MSI is insufficient to inform about its immunological phenotype and to predict response to immune therapy. With the advent of ICB as a powerful modality to treat advanced-stage MSI cancer patients, however lacking predictors of therapy success among MSI cancer patients, systematic characterization of (1) hereditary or sporadic origin, (2) immune infiltration and (3) immune evasion phenomena has become paramount in order to better delineate the determinants of therapy response. In particular, literature data suggesting differences with regard to immune evasion phenotypes between hereditary and sporadic MSI CRCs may point towards differential responsiveness to ICB. Although potentially being associated with additional costs and logistical hurdles, it is strongly encouraged to account for germline variant status and tumor immune status in future clinical studies addressing ICB in MSI cancer patients.

The findings of our study also have implications on tumor prevention. If immune surveillance governs MSI cancer evolution, modulation of immune surveillance may well hold the key towards effective prevention of MSI tumors, possibly by vaccination with shared MSI-induced FSP neoantigens.

## Figures and Tables

**Figure 1 jcm-09-01741-f001:**
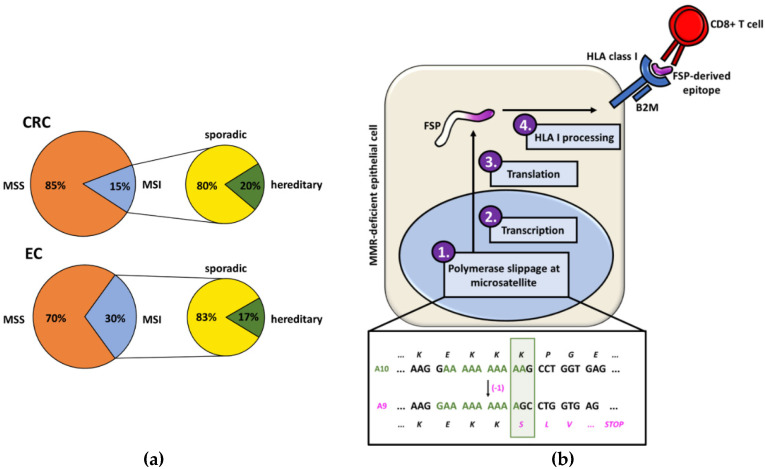
(**a**) Proportions of colorectal cancers (CRCs) and endometrial cancers (ECs) that present the microsatellite instability (MSI) phenotype and the respective percentages of MSI tumors with a hereditary background. (**b**) The generation of frameshift peptides (FSPs) starts with a polymerase slippage event in a microsatellite sequence that results in an insertion/deletion mutation. Due to the impaired mismatch repair (MMR) system, this error is not corrected and leads to a shift of the reading frame. After translation, the FSP is processed by the human leukocyte antigen (HLA) processing and presenting machinery (depicted: HLA class I) and FSP-derived neoantigens are presented on the surface of the MSI cell. Interaction of the HLA-bound antigen can trigger FSP-specific immune responses. The shown example is based on a mutation in the A10 microsatellite of the *TGFBR2* gene.

**Figure 2 jcm-09-01741-f002:**
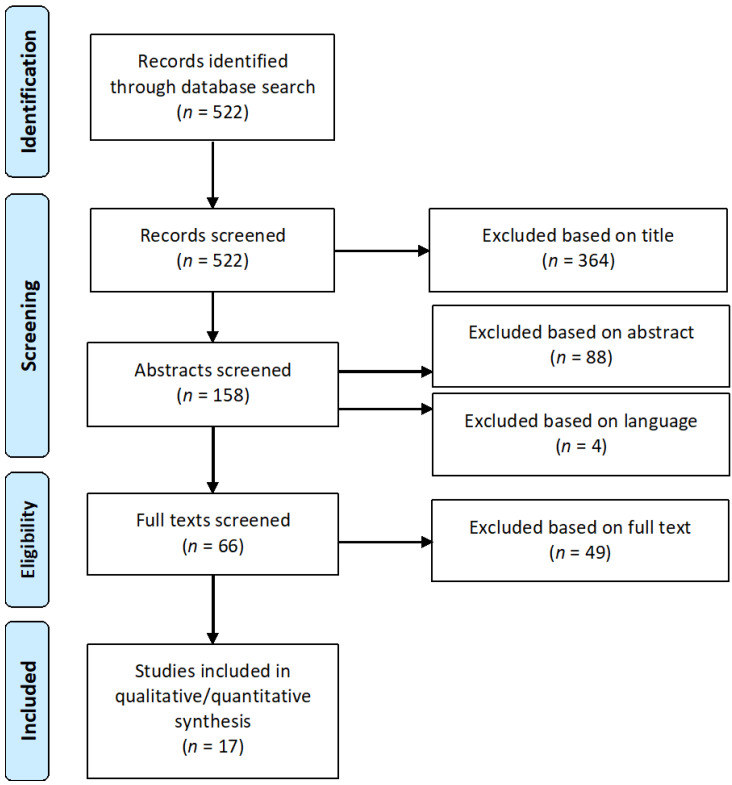
Flow diagram illustrating the systematic literature search according to Preferred Reporting Items for Systematic Reviews and Meta-Analyses (PRISMA) recommendations. The selection process identified 17 studies that were included in the analysis.

**Figure 3 jcm-09-01741-f003:**
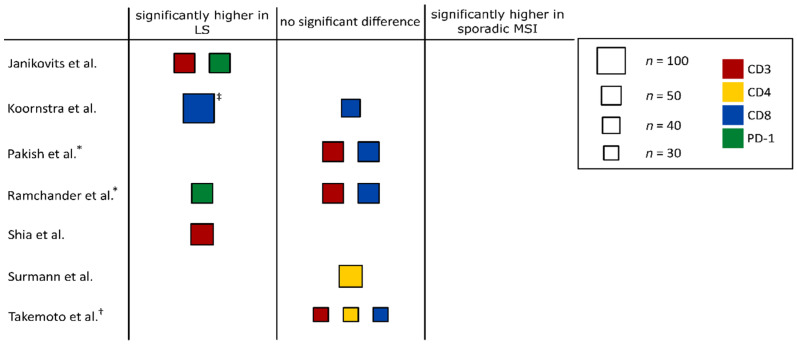
Graphical summary of the immune infiltration data in hereditary and sporadic MSI tumors. Studies that were assessed quantitatively were divided into three groups: studies reporting significantly higher immune infiltration counts in LS-associated MSI tumors compared to sporadic MSI tumors (“significantly higher in LS”), studies finding no significant differences between hereditary and sporadic MSI tumors (“no significant difference”) and studies reporting significantly higher immune infiltration counts in sporadic MSI tumors (“significantly higher in sporadic MSI”). Study size is proportional to the box area, and color indicates the respective marker. † study only counted immune cells in stromal regions. ‡ data for colorectal adenomas. * studies that analyzed MSI ECs.

**Figure 4 jcm-09-01741-f004:**
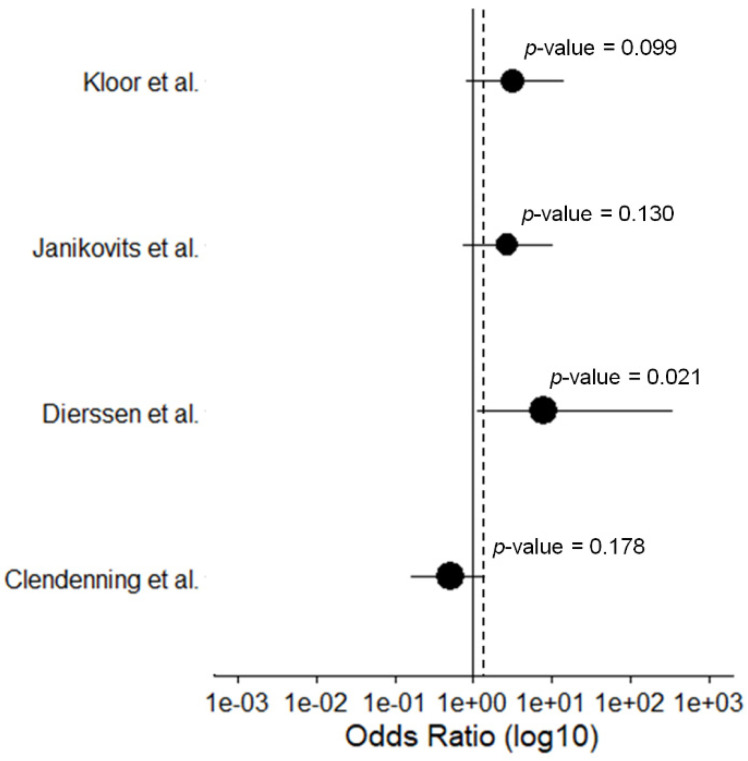
Forest plot of the calculated Odds Ratios for the *B2M* mutation frequency for hereditary vs. sporadic MSI CRCs with a 95% CI. Datapoints are proportional to study size. The dashed line indicates the total Odds Ratio.

**Figure 5 jcm-09-01741-f005:**
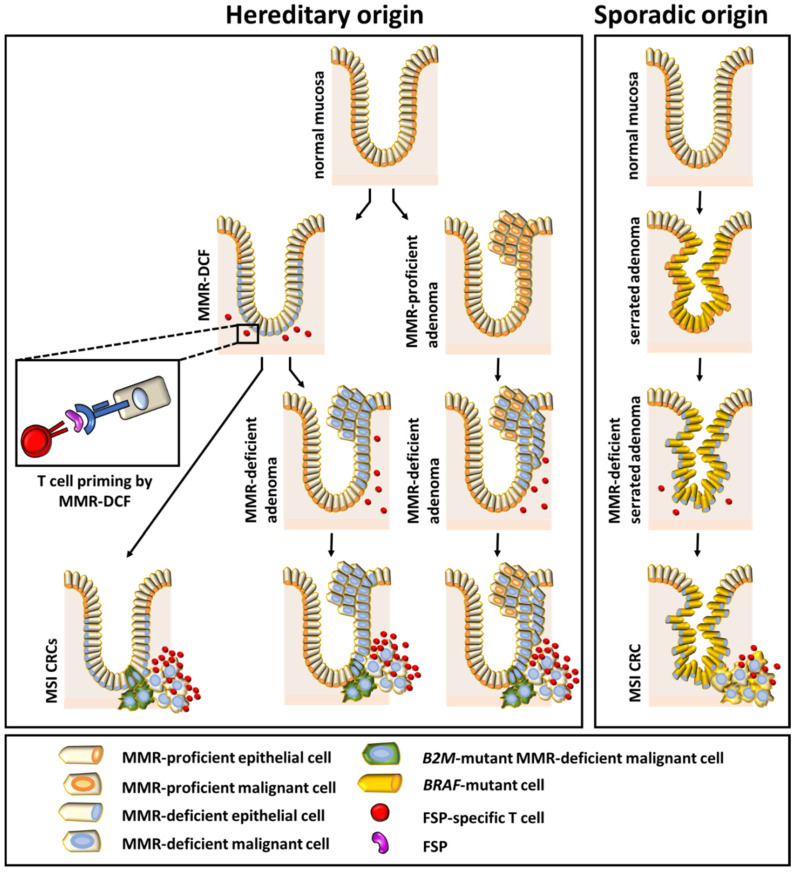
Summary figure showing the relationship between different colonic precursor lesions and the immune phenotype of the manifest tumor.

**Table 1 jcm-09-01741-t001:** Summary of papers quantitatively evaluated for T cell immune infiltration in hereditary and sporadic MSI CRCs, colorectal adenomas and ECs. The number of patient samples, LS status, applied T cell markers and quantification results are displayed; the respective counting methods are indicated. The T cell counts are given separately for tumor epithelium and stroma. Asterisk represents statistically significant differences between the hereditary and sporadic group. hpf = high-power-field.

		Sample Number		LS Status		Marker (Clone)	Positive Cell Counts		Analyzed Area	Counting Method
				Proven	Suspected					
**colorectal cancer**		Hereditary MSI CRCs	Sporadic MSI CRCs				Hereditary MSI CRCs	Sporadic MSI CRCs		
	**Janikovits et al., 2018 [47]**	*n* = 18	*n* = 38	x		CD3 (PS1)	143.1 *	92.5	tumor epithelial	median of positive cells per 0.25 mm^2^
						PD-1 (NAT105)	31 *	2.7		
	**Koornstra et al., 2009 [50]**	*n* = 20	*n* = 26	x	x	CD8 (na)	18.6	23.9	tumor epithelial	mean of positive cells in 10 hpf (400 × magnification)
	**Shia et al., 2003 [48]**	*n* = 30	*n* = 35		x	CD3 (na)	65 *	19	tumor epithelial	median of positive cells in 10 hpf (=1.96 mm^2^)
	**Surmann et al., 2015 [51]**	*n* = 35	*n* = 34	n.a.		CD4 (IF6)	53.4	68.9	tumor epithelial	median of positive cells per 0.25 mm^2^
						CD4 (IF6)	119.4	141.2	stromal	
	**Takemoto et al., 2004 [49]**	*n* = 19	*n* = 12		x	CD3 (×100)	165.4	153.2	stromal	mean/median of positive cells per 250 µm^2^
						CD4 (×20)	91.2	84.7		
						CD8 (×20)	51.5	47.6		
							no numbers for epithelial cells			
**colorectal adenomas**	**Koornstra et al., 2009 [50]**	**Hereditary Colorectal Adenomas**	**Sporadic Colorectal Adenomas**				**Hereditary Colorectal Adenomas**	**Sporadic Colorectal Adenomas**		
		*n* = 50	*n* = 69	x		CD8 (na)	6.6 *	3.9	epithelial	mean of positive cells in 10 hpf (400 × magnification)
**endometrial cancer**		**hereditary MSI ECs**	**sporadic MSI ECs**				**hereditary MSI ECs**	**sporadic MSI ECs**		
	**Pakish et al., 2017 [52]**	*n* = 20	*n* = 38		x	CD3 (SP7)	84.3	84.6	stromal	median of positive cells per 1 mm^2^
						CD4 (4B12)	16.3	22.1		
						CD8 (4B11)	82.8 *	34.2		
						PD-L1 (E1L3N)	289.3	297.9		
						CD3 (SP7)	18.7	30	tumor epithelial	
						CD4 (4B12)	2.8	8.6		
						CD8 (4B11)	8.2	4.2		
						PD-L1 (E1L3N)	4.8	8.2		
	**Ramchander et al., 2020 [53]**	*n* = 25	*n* = 33	x		CD3 (F7.2.38)	590	617	tumor center	mean of positive cells per 200 µm^2^
						CD8 (C8/144B)	291	233		
						CD45RO (UCLH1)	597	653		
						FoxP3 (236A/E7)	72	58		
						PD-1 (NAT105)	156 *	108		
						CD3 (F7.2.38)	386	241	invasive margin	
						CD8 (C8/144B)	287 *	116		
						CD45RO (UCLH1)	548 *	296		
						FoxP3 (236A/E7)	61	28		
						PD-1 (NAT105)	118 *	49		

**Table 2 jcm-09-01741-t002:** Summary of publications analyzing immune evasion in LS-associated and sporadic MSI CRCs and adenomas, distinguishing between mutations of the HLA class I and HLA class II pathways. Numbers of used samples are indicated for all groups and percentages of mutated samples are shown. Asterisk represents statistically significant differences between the hereditary and sporadic group.

	HLA I-*B2M* Mutations		Colorectal MSI Adenomas		LS Status	
	Hereditary MSI CRCs	Sporadic MSI CRCs			Proven	Suspected
Clendenning et al., 2018 [72]	17.1% (7/41)	29% (20/69)	0% (0/42)		x	x
Dierssen et al., 2006 [73]	20% (15/75)	3% (1/33)	x		x	
Janikovits et al., 2018 [47]	50% (9/18)	26.3% (10/38)	x		x	
Kloor et al., 2007 [12]	36.4% (16/44)	15.4% (4/26)	15.8% (6/38)		x	
Total	35.9% (47/131)	26.7% (35/131)	7.5% (6/80)			
95% CI	0.2816 to 0.4440	0.1985 to 0.3491				
*p*-value	0.1425					
	**HLA II-Mutations in *CIITA* and *RFX5***					
	***CIITA***		***RFX5***			
	**Hereditary MSI CRCs**	**Sporadic MSI CRCs**	**Hereditary MSI CRCs**	**Sporadic MSI CRCs**		
Surmann et al., 2015 [51]	8.9% (3/35)	0% (0/34)	14.3% (5/35)	26.5% (9/34)	n.a.	

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
