# Peer review of "Implications of Hereditary Origin on the Immune Phenotype of Mismatch Repair-Deficient Cancers: Systematic Literature Review"

_jcm, 2020, doi:10.3390/jcm9061741_

Round 1
Reviewer 1 Report
Summary:
The manuscript entitled “Implications of hereditary origin on the immune phenotype of mismatch repair-deficient cancers: Systematic literature review” is a well written precis
of the current literature and data detailing the immune landscape of sporadic MMRd and LS associated cancers. The authors are known experts in the area and well placed to summarise the current evidence base.
This systematic review provides clear evidence that LS associated cancers immunologically distinct from sporadic MMRd cancers. This is important as we move towards immune check point inhibition in the treatment of MMRd cancers. The current evidence base comes from studies that had high reliance on LS patients (Le et al 2015 and Le et al 2017). This is at odds with the clinical picture in which sporadic MMRd cancers are more common. If there are no immunological differences between these populations than this is of little relevance. However, if (as the authors demonstrate) there is a difference it may be that studies exploring immune therapies explore the impact of tumour aetiology on treatment outcomes. Therefore, this work should help inform future trail design and our general understanding of role of immunity and carcinogenesis in MMRd.
The authors should be commended on their work. I have made some suggestions as detailed below.
Data checks:
I completed some spot checks on the data presented in table one and have found no errors in their data extraction
Missing literature:
I have ran a few quick literature searches and have found no manuscripts that seem to have been missed. I have not done a systematic review of my own.
I know of no other literature that should be included.
Edits required (please act on these)
The following sentence:
“Notably, treatments supporting the anti-tumoral immune response, such as the recently developed immune checkpoint blockade (ICB) therapy showed great success selectively in patients with metastasized MSI cancers”
Should read:
“Notably, treatments supporting the anti-tumoral immune response, such as the recently developed immune checkpoint blockade (ICB) therapy showed great success specifically in patients with metastasized MSI cancers”
The following sentence:
“LS is caused by a pathogenic variant in one of the MMR genes, and 62 LS-associated tumors develop though the inactivation of the MMR system upon a subsequent 63 somatic hit in the second functioning allele”
Needs to be amended so as to include the rarer causes of LS such as path_EPCAM or inherited MLH1 hypermethylation. For example: LS is caused by an inherited inactivation of the MMR genes
I could not see figure 1 due to an editing issue.
Regarding the general layout of the manuscript, I feel a few more heading titles would help the reader navigate the text. Suggested additional titles:
Immune infiltration in hereditary and sporadic MSI non CRC cancers
Immune infiltration in hereditary and sporadic MSI pre-malignant lesions
The summary at line 255 is in an odd place as it seems to summarise a section before it is complete – that is it is in the middle of a topic.
Regarding the search methodology:
- You do not search pubmed, you search the database Medline. Pubmed is the interface software.
- Did you search other databases such as EMBASE or Web of Science etc?
- Can you included the dates of the search – something akin to “databases were search from the date of their conception to 15.5.20”
- Were MeSH or Emtree terms used? If not why not?
- Did you do citation searching in those manuscripts you did full paper reviews?
- Did you search the Grey literature?
- How did you ensure duplicates were removed?
- What were your inclusion/exclusion criteria – these need to be clearly stated.
- Please state the software/method used to screen the abstracts
- Did two independent reviews screen the titles? Who made the decision if these reviewers could not agree? Duplicate review is best practice: https://doi.org/10.1002/jrsm.1369 therefore if this was not performed it must be stated and I would suggest a subset of the original search results is checked by a second reviewer with a kappa coeffect calculated to ensure the first review would have been consistent with a second reviewer.
- It is best practice to present all the papers that were excluded at the point of full manuscript review in a table with reasons as to why these papers were excluded. Please could you provide this in a supplementary table? This is in keeping with the PRISMA recommendations and considered best practice.
The statistical methods used in this paper are not described in the methods. Please could you add them along with the software package used.
The way in which Lynch syndrome is defined by the authors of the included studies is of vital importance. For example, Pakish et al the following is used a definition of LS:
“ Lack of protein expression of MSH2, MSH6, or PMS2 in the tumor was considered probable Lynch syndrome. For those with MLH1 loss by IHC, MLH1 promoter methylation was analyzed. Those cases showing MLH1 loss by IHC and without MLH1 promoter methylation were classified as probable Lynch syndrome.”
This does not exclude somatic path_MMR and therefore may include non-LS cancers in their LS cohort. In contrast, Ramchander et al all cases had to have proven germline mutation. I would add a column to table one with Proven or Suspected LS to help the reader interpret the data. I would also discuss this in the review as a means of confounding within your results
In Table one please add the study citations numbers after the authors names so that they align with your list of references.
After the sentence: “Seven studies were informative for quantitatively evaluating the extent of local immune 186 infiltration in MSI CRCs (Table 1).” You should cite the seven studies.
Regarding the following: “Further, a study by Jass 196 et al. (2002), which was identified independently of the literature search, also mentions a higher 197 frequency of Crohn’s like reactions and peritumoral T cells in hereditary compared to sporadic MSI 198 CRCs [47].” Could you explain how this study was identified?
The following sentence:
“In summary, the existing literature provides ample evidence that LS-associated MSI tumors generally have a more pronounced local T cell infiltration compared to sporadic MSI CRCs (Figure 226 3), also referring to T cells expressing markers for cytotoxicity and activation.” does not make sense. Please could you rewrite?
Figure 4 – it is not clear what method of analysis was used to generate these data.
In the sentence:
“In addition, two studies analyzed the frequency of B2M mutations in adenomas: Kloor et al. 327 (2007) reported a 15.8% B2M mutation frequency in 38 MSI adenomas [12]. In contrast, Clendenning 328 et al. (2018) did not observe any B2M mutations in 42 MMR-deficient adenomas [69].”
Are these MSI sporadic/LS or both? Please ensure your nomenclature is consistent throughout to ensure there is not confusion around these terms.
Please define the abbreviation UICC
The following: “As the activity of the immune system and probably its capacity to recognize tumor cells is related to age [83–85], one may speculate that the observed elevated immune responses in LS mutation carriers compared to sporadic MSI tumor patients may merely reflect the generally younger”
Should read
As the activity of the immune system and probably its capacity to recognize tumor cells are related to age [83–85], one may speculate that the observed elevated immune responses in LS mutation carriers compared to sporadic MSI tumor patients may merely reflect the generally younger”
The following statement :
“Our study demonstrates that classification of a tumor as MSI is insufficient to inform about its immunological phenotype and to predict response to immune therapy. With the advent of ICB as a powerful modality to treat advanced-stage MSI cancer patients, systematic characterization of (1) hereditary or sporadic origin, (2) immune infiltration, and (3) immune evasion phenomena has become paramount.”
Could the authors expand a little more as to why it is paramount. Treatment with ICB has been licenced on the basis of MSI status not on the level of immune response in the tumour. It is not common clinical practice to further define the immune landscape and treatments are started without such work. So why is it paramount? What do the authors think this will add? Is this in the context of future trials or do the authors want this to take place in clinical practice – something that would be time consuming and expensive so it would needs to be justified.
Suggestions: (may be acted on)
The following sentence: “We found evidence that recurrent encounters with premalignant lesions may raise the alertness of the immune system against MSI-associated tumor antigens in LS individuals (Figure 5).” You may wish to use more recognised language such as “primes the immune system”
The authors may wish to develop a short discussion around the impact of the host’s immune system on the oncological immune response. If so this article is of interest.
DOI: 10.1200/JCO.2019.37.15_suppl.1532 Journal of Clinical Oncology 37, no. 15_suppl (May 20, 2019) 1532-1532.
In addition, it would be interesting to develop the discussion around B2M expression and metastasis. That is do you need to express B2M to enable metastasis? Could this explain the improved survival?
Reviewer 2 Report
I read with great interest your work entitled "Implications of hereditary origin on the immune phenotype of mismatch repair-deficient cancers: Systematic literature review".
MMR deficiency, leading to MSI development, can occur sporadically or is associated with Lynch syndrome. At the beginning of the abstract, the Authors mention both colorectal cancer and endometrial cancer, as MSIs are most common in these cancers. Moreover, LS is associated with a high risk of CRC, as well as EC (second most common). Later in the abstract, the Authors state that their systematic literature search was performed to collect data on the differences between sporadic and LS-associated tumors regarding local T cell infiltration and immune evasion mechanisms, which, based on what has been written so far, may suggest that the Authors will focus on CRC and EC.
The introduction focuses primarily on colorectal cancer. The Authors point out that no analysis that compares the immune characteristics of LS and sporadic MSI tumors has been performed so far and therefore, this review will focus on this issue. The methodology section is presented in a clear way. In the case of results, CRC related aspects are described in great detail, whereas the EC part is relatively short, especially when compared to CRC.
In conclusion, the work is interesting, and the Authors' commitment to the topic is visible.
My recommendations/questions mainly concern organization and clarification of several aspects:
1. The abstract states that the analysis was performed for LS-associated tumors, so it would be worth specifying that it was done for CRC and EC, with particular focus on CRC.
2. I recommend to consider whether the current title is not too general.
3. In the Introduction, the Authors stated that the analysis was performed for MSI colorectal cancers, but it would be worth adding that EC was also taken into account as it is shown later in the results.
4. In the results section 3.2:
a) it would be worth to include ECs in addition to sporadic MSI CRCs in the title
b) ECs should be included in the Table 1 name
c) it is worth to add the publication year or reference number next to the name of the quoted author in Table 1.
5. The greater part of the article focuses on the CRC, which is why I wanted to ask the Authors whether it was intentional, or is it due to less information about EC, which may result from e.g. excluding some papers during analysis or the lack of experimental studies? If the latter, then it is worth mentioning in the article.
6. The Authors stated that patients with MSI CRCs generally show more favorable prognosis compared to patients with MSS tumors. Is it affected by cancer stage?
7. The authors use the term "MSI cancers" in the summary. Does this mean that your findings relate to these cancers as a group? In this case, will the conclusions drawn based on the work related to CRC and EC apply to other MSI cancers, e.g. gastric or ovarian cancer?
